# Resilience of small intestinal beneficial bacteria to the toxicity of soybean oil fatty acids

Sara C Di Rienzi[1,2], Juliet Jacobson[2], Elizabeth A Kennedy[2], Mary E Bell[2], Qiaojuan Shi[2], Jillian L Waters[1,2], Peter Lawrence[3], J Thomas Brenna[3,4], Robert A Britton[5], Jens Walter[6,7], Ruth E Ley[1,2]*

[1]Department of Microbiome Science, Max Planck Institute for Developmental Biology, Tübingen, Germany; [2]Department of Molecular Biology and Genetics, Cornell University, Ithaca, United States; [3]Division of Nutritional Sciences, Cornell University, Ithaca, United States; [4]Dell Pediatric Research Institute, Dell Medical School, University of Texas at Austin, Austin, United States; [5]Department of Molecular Virology and Microbiology, Baylor College of Medicine, Houston, United States; [6]Department of Agricultural, Food, and Nutritional Science, University of Alberta, Edmonton, Canada; [7]Department of Biological Sciences, University of Alberta, Edmonton, Canada

**Abstract** Over the past century, soybean oil (SBO) consumption in the United States increased dramatically. The main SBO fatty acid, linoleic acid (18:2), inhibits in vitro the growth of lactobacilli, beneficial members of the small intestinal microbiota. Human-associated lactobacilli have declined in prevalence in Western microbiomes, but how dietary changes may have impacted their ecology is unclear. Here, we compared the in vitro and in vivo effects of 18:2 on *Lactobacillus reuteri* and *L. johnsonii*. Directed evolution in vitro in both species led to strong 18:2 resistance with mutations in genes for lipid biosynthesis, acid stress, and the cell membrane or wall. Small-intestinal *Lactobacillus* populations in mice were unaffected by chronic and acute 18:2 exposure, yet harbored both 18:2- sensitive and resistant strains. This work shows that extant small intestinal lactobacilli are protected from toxic dietary components via the gut environment as well as their own capacity to evolve resistance.

DOI: https://doi.org/10.7554/eLife.32581.001

*For correspondence: ruth.ley@tuebingen.mpg.de

## Introduction

While antibiotics can cause lasting alterations to the microbiome (*David et al., 2014a*; *Dethlefsen et al., 2008*; *Dethlefsen and Relman, 2011*; *Jakobsson et al., 2010*), dietary perturbations rarely do so (*Sonnenburg et al., 2016*). In humans and in mice, the gut microbiome can be quickly altered by diet but community composition generally recovers within days (*Carmody et al., 2015*; *David et al., 2014a*; *David et al., 2014b*; *Zhang et al., 2012*). Resilience to dietary perturbation may be direct, as gut microbes functionally adapt to diet, or indirect through buffering by the gut habitat.

During the 20[th] century, the greatest dietary change in the United States was in the consumption of soybean oil (SBO), which increased from less than 0.001 kg/person/year to 12 kg/person/year (*Blasbalg et al., 2011*). Conventional ('commodity') soybean oil, frequently labeled as 'vegetable oil', is a mixture of triglycerides composed of five long chain fatty acids (FAs), with linoleic acid (18:2) comprising over 50% of the FAs. After triglycerides are hydrolyzed by lipases active in the saliva, stomach, and upper duodenum, free FAs and monoglycerides are absorbed in the small intestine

**eLife digest** Though "you are what you eat" may just be a figure of speech, it is clear that what we eat does affect our own cells and the microbes that live in our gut. During the 20th century, the American diet changed dramatically and now includes a lot more vegetable oil from soybeans, the main component of which – a fat called linoleic acid – is toxic to many microbes.

Among the microbes inhibited by linoleic acid is a beneficial bacterium called *Lactobacillus reuteri*. This microbe has become less common in Western populations, and the timing of its decline approximately follows when the consumption of soybean oil began increasing. However, *L. reuteri* and other related microbes still exist in people who eat a Western diet. This suggests that these bacteria must be protected from linoleic acid in the gut, or that they can become resistant to this toxic molecule.

Now, Di Rienzi et al. report that both protection in the gut and resistance could explain how *L. reuteri* can persist in the presence of linoleic acid. First, experiments in the laboratory showed that these microbes could indeed become resistant to linoleic acid, either by gaining mutations in genes involved in creating fats, by growing in an acid, and by forming a cell wall. Further experiments involving mice then showed that the gut protects also *L. reuteri* from this molecule: linoleic acid did not inhibit *L. reuteri* within the mouse, but those same *L. reuteri* were inhibited when grown outside of a mouse.

Di Rienzi et al. went on to recover some resistant *L. reuteri* from mice, implying that there is a mix of resistant and non-resistant strains in the mouse gut. However, notably, the resistant bacteria recovered from the mice did not have mutations in the genes that had been identified from the earlier experiments.

Together these findings show that gut bacteria have several means of surviving the high levels of potentially toxic fat molecules. Also, the specific finding that linoleic acid does not inhibit *L. reuteri* within the gut may help scientists to understand how a high fat diet affects microbes; for example, it is possible that the decrease in carbohydrates or protein that occurs in high fat diets may explain why such diets cause microbes to be lost. Lastly, and on a practical level, linoleic acid-resistant *L. reuteri* may in the future be used as a probiotic in foods rich in vegetable oil.

DOI: https://doi.org/10.7554/eLife.32581.002

(*Mansbach et al., 2000*). The microbiota of the human small intestine is exposed to FAs during this process (*El Aidy et al., 2015*; *Kishino et al., 2013*): therefore, an increase in the concentration of specific FAs has the potential to reshape microbial communities and select for microbes that thrive in the novel environment.

Linoleic acid and the other two major unsaturated FAs in SBO, oleic acid (18:1), and alpha-linolenic acid (18:3), are known to be bacteriostatic and/or bactericidal to small intestinal bacteria as non-esterified (free) fatty acids in vitro at concentrations found in the small intestine (*Kabara et al., 1972*; *Kankaanpää et al., 2001*; *Kodicek, 1945*; *Nieman, 1954*). The primary modes of killing include permeabilization of cell membranes (*Greenway and Dyke, 1979*) and interference with FA metabolism (*Zheng et al., 2005*). Affected microbes are predominantly Gram-positive bacteria including the genus *Lactobacillus* (*Nieman, 1954*). Lactobacilli are particularly important as they are considered beneficial members of the human small intestine (*Walsh et al., 2008*; *Walter et al., 2007*; *Walter et al., 2011*). They have been shown to be growth inhibited by the specific FAs present in SBO (*Boyaval et al., 1995*; *De Weirdt et al., 2013*; *Jenkins and Courtney, 2003*; *Jiang et al., 1998*; *Kabara et al., 1972*; *Kankaanpää et al., 2001*; *Kodicek, 1945*; *Raychowdhury et al., 1985*). It is interesting to note that the human-associated *L. reuteri* underwent a population bottleneck that coincides with the increase in SBO consumption in the U.S. and is far less prevalent than it was in the past (*Walter et al., 2011*). Despite its decline, *L. reuteri* and other lactobacilli persist in the small intestine of Western individuals, suggesting mechanisms to counter the inhibitory effects of FAs in vivo.

Here, we explored mechanisms of microbiome resistance to toxic dietary components with a focus on linoleic acid (18:2) toxicity to *L. reuteri* and *L. johnsonii*. Using an in vitro evolution assay, we assessed the capacity for these species to develop 18:2 resistance. To assess resistance in the

host, we fed mice from two vendors diets high or low in 18:2 for 10 weeks and exposed their intestinal microbes to acute dosing of 18:2 via gavage. Lactobacilli populations were quantified in live-only and whole cell fractions obtained from the small intestine, and isolates from mice were assessed for resistance in vitro.

## Results

### *L. reuteri* strains show variable resistance to 18:2 in vitro

We confirmed the previously reported in vitro toxicity of long chain FAs towards *L. reuteri* by performing disc diffusion assays with the individual free FAs of soybean oil (SBO) using *L. reuteri* ATCC 53608. We observed growth inhibition of this strain by free 18:1, 18:2, and 18:3 (*Figure 1A*), and this inhibition occurred in the presence of completely hydrolyzed SBO (*Figure 1—figure supplement 1*). The two saturated free FAs 16:0 and 18:0 and glycerol did not interfere with growth. To determine if the inhibitory concentration of 18:2 was comparable to concentrations in the mammalian digestive tract, we performed a cell permeability assay using propidium iodide with *L. reuteri* ATCC 53608 over a 10-fold dilution range from 0.01 to 1000 µg/ml of 18:2. We observed that 18:2 permeabilized the cells with an estimated inhibitory concentration 50 (IC50) of 20 µg/ml ($p < 0.001$) (*Figure 1B*). This IC50 concurs with our estimates of the concentration of 18:2 present in a mouse consuming a SBO diet (11 to 28 µg/ml for a mouse on a 7% by weight SBO diet, see Materials and methods) and with previous estimates of mammalian physiological relevant concentrations of unsaturated FAs (*Kankaanpää et al., 2001*; *Kodicek, 1945*). Thus, physiological levels of 18:2 were toxic to *L. reuteri in vitro*.

We next assessed 40 strains of *L. reuteri* for 18:2 resistance in liquid culture. These 40 strains were previously isolated from humans, pigs, rodents (mice, rats), birds (chicken, turkey), and sourdough and stemmed from six different continents (*Supplementary file 1*) (*Böcker et al., 1995*; *Oh et al., 2010*). We quantified how the strains grew in 18:2 by taking the mean of the ratios for cells growing in 18:2 to cells growing in medium alone for each of the last three $OD_{600}$ measurements at hours ~ 4, 6, and 8 during the growth assay (see Materials and methods for a discussion on why this approach was used). *L. reuteri* strains have been shown to be host-specific and form host-specific clades (*Walter et al., 2011*). We observed that the basal, rodent-associated strains on average were inhibited by 18:2 more strongly than the other strains (Kruskal-Wallis test, $p < 10^{-4}$) (*Figure 2*). However, we observed considerable variation within host sources, and the human-associated

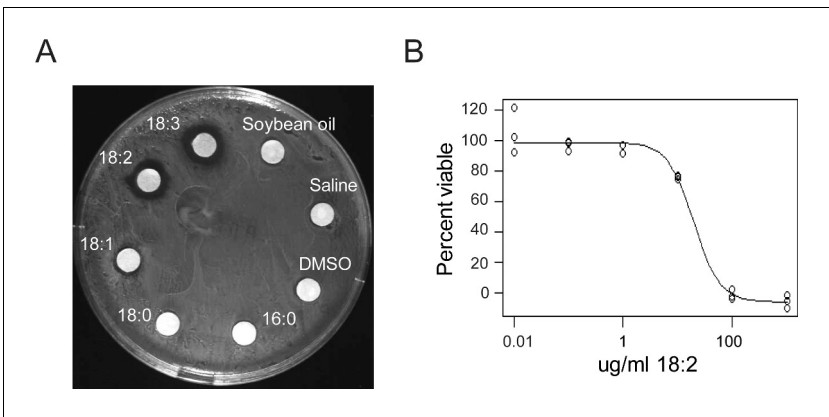

**Figure 1.** *L. reuteri* is inhibited by 18:2. (**A**) Disc diffusion of *L. reuteri* plated with the FAs of SBO. FAs were dissolved in DMSO to a concentration of 50 mg/ml, except for palmitic acid (16:0), which was dissolved to a concentration of 5 mg/ml. Clearings around the discs indicate growth inhibition. (**B**) Dose response curve of 18:2 with *L. reuteri*. IC50 is estimated at 20 µg/ml ($p < 0.001$). See also *Figure 1—figure supplement 1*.
DOI: https://doi.org/10.7554/eLife.32581.003
The following figure supplement is available for figure 1:

**Figure supplement 1.** *L. reuteri* is inhibited by the hydrolysis products of SBO.
DOI: https://doi.org/10.7554/eLife.32581.004

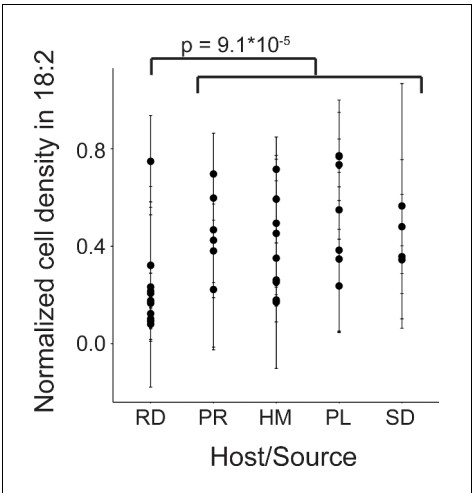

**Figure 2.** Variation in natural *L. reuteri* strains' response to 18:2. Fourteen rodent (RD), six porcine (PR), nine human (HM), seven poultry (PL), and four sourdough (SD) strains were tested. Standard deviations in normalized cell density in 18:2 are shown. Higher values indicate cellular density achieved in liquid culture is uninhibited by 18:2. Significance was determined by a Kruskal-Wallis test; mean in non-rodents = 0.45, mean in rodents = 0.22. See also *Figure 2—figure supplement 1* and *Supplementary file 1*.

DOI: https://doi.org/10.7554/eLife.32581.005

The following figure supplement is available for figure 2:

**Figure supplement 1.** *L. reuteri* resistance to 18:2 is not related to site of isolation in humans nor phylogenetic clade.

DOI: https://doi.org/10.7554/eLife.32581.006

strains were no more resistant to 18:2 than the strains derived from pig, poultry, or sourdough. Moreover, within human strains, 18:2 resistance did not relate to *L. reuteri* isolation site (*Figure 2—figure supplement 1A*). There also did not appear to be a clear relationship of 18:2 resistance with *L. reuteri* clades as defined by *Oh et al., 2010* (*Figure 2—figure supplement 1B*). Overall, we observed variation in *L. reuteri* 18:2 resistance regardless of source.

## Evolved 18:2 resistance is associated with mutations in lipid-related, acid stress, and cell membrane/wall genes

To directly test if 18:2 resistance could evolve in *L. reuteri* through exposure to 18:2, we isolated an 18:2-sensitive *L. reuteri* strain (LR0) from the jejunum of a conventionally-raised mouse (see Materials and methods). We seeded five cultures with LR0 and passaged them twice daily from a growth-dampening concentration of 18:2 up to a growth-inhibitory concentration over a period of six weeks (*Figure 3A* and *Figure 3—figure supplement 1*). We also evolved five cultures of *L. johnsonii* strain (LJ0) obtained from the same mouse. We selected *L. johnsonii* based on its high abundance in mouse small intestinal microbiota (see below). At the end of the passaging regime, all of the evolved lactobacilli populations showed smaller zones of inhibition around 18:2 and 18:3 in a disc diffusion assay compared to their respective starting strains (*Figure 3B* and *Figure 3—figure supplement 2A*). We tested isolates LR2-1 from population LR2 and LJ41072 from population LJ4 in liquid culture supplemented with 18:2 to confirm their 18:2 resistance (*Figure 3C and D*).

To characterize the mutations these populations acquired, we sequenced all five of the *L. reuteri* populations, four of the five *L. johnsonii* populations (the fifth was lost), the evolved isolates LR2-1 and LJ4107, and the starting strains LR0 and LJ0, using 300 bp paired end sequencing on an Illumina MiSeq. For the populations, we achieved approximately 500X coverage, and for the isolates, 50X coverage (*Supplementary file 2*). Mutations were called in the populations and isolates by aligning sequencing reads to the assembled genome for the respective starting strain (LR0 or LJ0). After requiring mutations have a minimum frequency of 10% in a population and confirming all mutations were not due to potential mismapping, we observed 30 mutational events in 15 genes across the five *L. reuteri* populations and 35 mutational events in 21 genes in the four *L. johnsonii* populations. (*Supplementary file 3* and *4*).

In each population, a few mutations had swept the entire population (*Tables 1* and *2*, *Supplementary file 3* and *4*). Both the *L. reuteri* and *L. johnsonii* populations bore high frequency variants (>60%) in genes relating to FA metabolism, ion transport, and the cell membrane/wall. In the *L. reuteri* populations, we found high frequency variants in (i) FA biosynthesis transcriptional regulator *FabT* (*Eckhardt et al., 2013*), (ii) two related tyrosine-protein kinases involved in exopolysaccharide synthesis, *EpsD*, and *EpsC* (*Minic et al., 2007*), (iii) an HD family hydrolase, (iv) a hypothetical protein, and (v) in the region upstream of an ammonium transporter that may respond to acid stress (*Wall et al., 2007*). In the *L. johnsonii* populations, high frequency mutations were present in (i) two distinct intracellular lipases, (ii) a putative membrane protein gene, (iii) the

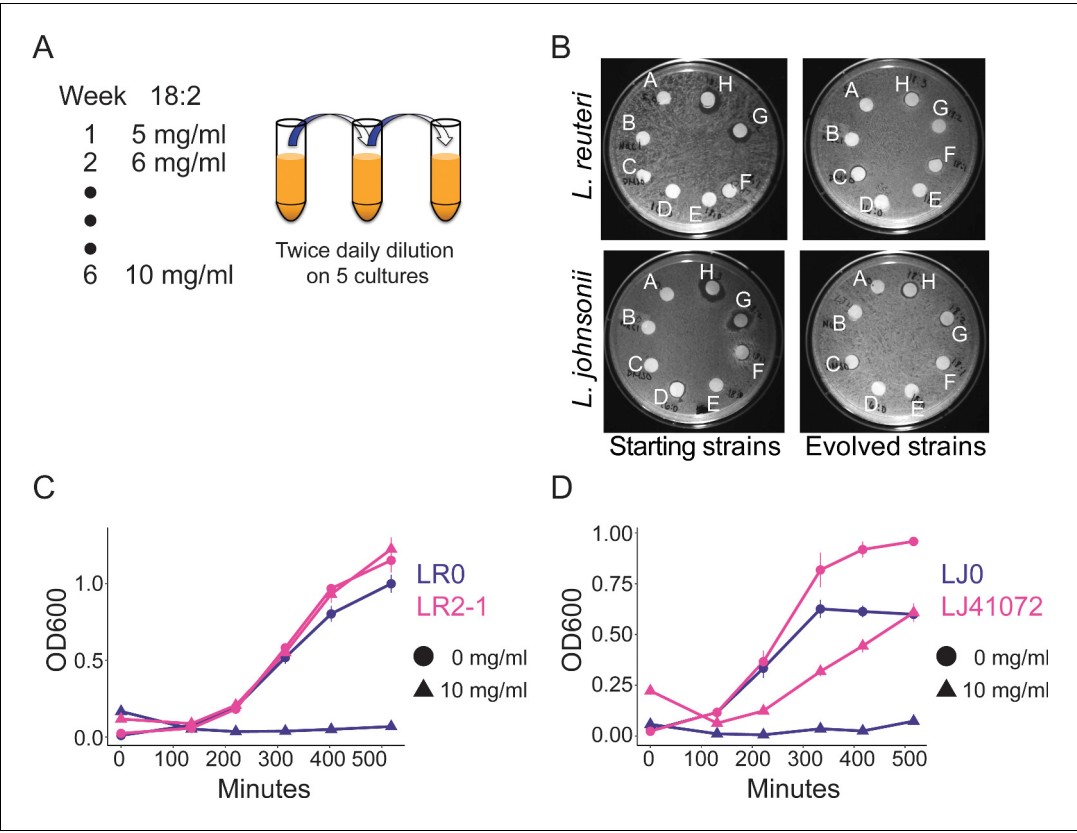

**Figure 3.** In vitro evolution of 18:2 resistance in lactobacilli. (**A**) Five cultures of *L. reuteri* strain LR0 and five cultures of *L. johnsonii* strain LJ0 were passaged twice daily via a 100x dilution in liquid culture supplemented with 18:2. The 18:2 concentration was increased each week by 1 mg/ml from 5 to 10 mg/ml over a total of 6 weeks. (**B**) Disc diffusion (as in *Figure 1*) of *L. reuteri* and *L. johnsonii* starting strains LR0 and LJ0 and evolved populations LR2 and LJ4. Tested compounds: A. SBO, B. Saline, C. DMSO, D. 16:0, E. 18:0, F. 18:1, G. 18:2, H. 18:3. Growth curve of **C**) *L. reuteri* starting strain LR0, evolved isolate LR2-1 (from population LR2), (**D**) *L. johnsonii* starting strain LJ0, and evolved isolate LJ41072 (from population LJ4) in liquid medium with and without 18:2. Each point represents triplicate cultures and standard deviations are shown. See also *Figure 3—figure supplements 1* and *2* and *Supplementary file 2 to 6*.

DOI: https://doi.org/10.7554/eLife.32581.007

The following source data and figure supplements are available for figure 3:

**Source data 1.** Oligos used to generate *L. reuteri* mutants.
DOI: https://doi.org/10.7554/eLife.32581.010

**Figure supplement 1.** *Lactobacillus* strains, populations, and isolates involved in the in vitro evolution experiment.
DOI: https://doi.org/10.7554/eLife.32581.008

**Figure supplement 2.** *Lactobacillus* populations passaged in 18:2 have increased resistance to 18:2.
DOI: https://doi.org/10.7554/eLife.32581.009

potassium efflux system *KefA*/small-conductance mechanosensitive channel, which protects against growth defects in acidic conditions (*Cui and Adler, 1996*; *McLaggan et al., 2002*), (iv) the glycosyl-transferase *LafA*, which affects the lipid content of the cell wall and membrane (*Webb et al., 2009*), (v) a *TetR* family transcriptional regulator, and (vi) the ribonucleotide reduction protein *NrdI*. All but two of the above mutations are non-synonymous or cause protein truncations. The other two mutations are intergenic and thus may alter the expression of the downstream gene. The isolate LR2-1 contained both of the mutations present at high frequencies in the total LR2 population as well as an additional mutation in a hypothetical protein, which was present in the LR2 population at 45% (*Supplementary file 3*). Similarly, LJ41072 had all of the high frequency mutations present in its source population (LJ4) and one additional mutation in LafA, which was mutated in 39% of the LJ4

**Table 1.** High frequency mutations in *L. reuteri in vitro* evolved populations.

| Gene | Function | LR1 | LR2 | LR3 | LR4 | LR5 |
|---|---|---|---|---|---|---|
| *FabT* (5)* | Fatty acid biosynthesis | 71% NS | 99% IT | 98% NS | 76% U | 81% NS |
| *EpsD* (2)* | Exopolysaccharide synthesis | 76% FS | | 99% NS | | |
| *EpsC* | Exopolysaccharide synthesis | | | | | 86% NS |
| FIG005986 HD family hydrolase | Hydrolase | | | | | 77% NS |
| FIG00745602 hypothetical protein | Transmembrane protein | | 99% PS | | | |
| Ammonium transporter | Ammonium transporter | | | 67% U | | |

*(#) indicates number of distinct mutations across the populations. The percent of the population with a mutation in the named gene is shown. Variants at frequency greater than 60% are shown. NS = nonsynonymous; IT = internal truncation; U = intergenic upstream; PS = premature stop; FS = frameshift. See also **Figure 3—figure supplements 1** and **2** and **Supplementary Files 3** and **5**.

DOI: https://doi.org/10.7554/eLife.32581.011

population (**Supplementary file 4**). We observed no overlap in the specific genes mutated in *L. reuteri* and *L. johnsonii.* Only a subset of the genes mutated in one species are present in the other species (*EpsD*, *EpsC*, FIG00745602, *LafA*) and in no case was the same mutation already present in the opposite species. Although the specific genes mutated differed between the two species, they are associated with similar functions, suggesting that *Lactobacillus* species can evolve 18:2 resistance through changes relating to lipid metabolism, acid stress, and the cell wall/membrane.

To confirm the role of these genes in fatty acid resistance, we generated these mutations individually in a fatty acid sensitive background. The human derived *L. reuteri* ATCC PTA 6475 (also called MM4-1A) is amenable to recombineering (**van Pijkeren and Britton, 2012**). Of the genes mutated in *L. reuteri*, only FabT and the hydrolase gene are present in this strain. The amino acid sequences, but not the nucleotide sequences of these genes are identical between our mouse strain and *L. reuteri* 6475. We created the LR2 18 bp deletion in FabT, the LR5 SNP in FabT, and the LR5 SNP in the hydrolase gene. The latter two were accompanied by several surrounding synonymous mutations as recombineering is orders of magnitude more efficient when multiple consecutive mutations are made due to the avoidance of the mismatch repair system (**van Pijkeren and Britton, 2012**). The specific mutations made are indicated in the recombineering oligos in **Figure 3—source data 1**. Note that these oligos match the reverse strand of the chromosome.

The LR2 18 bp deletion and the LR5 SNP in FabT present alone were able to enhance 18:2 resistance in *L. reuteri* 6475, similar to that observed for the total LR2 and LR5 populations (**Figure 3—figure supplement 2B**). The LR5 SNP in the hydrolase gene, however, was not sufficient to render the strain observably 18:2 resistant by a disc diffusion assay. We cannot rule out the possibility that the additional synonymous mutations we created in this strain impacted the phenotype or that mutation of the hydrolase gene enhances resistance in the background of a strain mutated for FabT. These results verify the role of the fatty acid transcriptional regulator FabT in *L. reuteri* 18:2 resistance.

**Table 2.** High frequency mutations in *L. johnsonii in vitro* evolved populations.

| Gene | Function | LJ2 | LJ3 | LJ4 | LJ5 |
|---|---|---|---|---|---|
| Esterase/lipase | Intracellular esterase/lipase | 88% NS | | | |
| Putative membrane protein (2)* | Transmembrane protein | 100% NS | 100% FS | 100% NS | |
| Lipase/esterase | Intracellular esterase/lipase | | 99% NS | | 93% NS |
| *KefA* | Small-conductance mechanosensitive channel | | | 62% DEL | |
| *LafA* | Glycosyltransferase | | 100% NS | | |
| *NrdI* (2)* | Ribonucleotide reduction | 100% NS | | 100% NS | |
| *TetR* family transcriptional regulator | Membrane structure | | | >60% PS | |

Data are presented as in **Table 1**. NS = nonsynonymous; FS = frameshift; DEL = in frame deletion; PS = premature stop. The insertion in *TetR* in LR4 was not properly called by GATK; the frequency is estimated. See also **Figure 3—figure supplements 1** and **2** and **Supplementary file 4** and **6**.

DOI: https://doi.org/10.7554/eLife.32581.012

## *L. reuteri* survives chronic and acute 18:2 exposure in the mouse

Given that 18:2 resistance can evolve in vitro, we asked if *L. reuteri* and *L. johnsonii* could survive either a chronic or acute exposure to 18:2 in vivo. For the chronic exposure, 3 week-old male C57BL/6J mice from Jackson Laboratories were fed *ad libitum* for 10 weeks a low fat (LF, 16% kcal from SBO) or high fat (HF, 44% kcal from SBO) diet, wherein all of the fat was derived from SBO (*Figure 4—source data 1*). For the acute exposure, at the end of the 10 weeks, we gavaged (delivered to the stomach) mice with 6 mg 18:2 per gram mouse weight (e.g., double the 18:2 consumed by mice daily on the LF diet) or saline. At 1.5 hr post-gavage, when gavaged 18:2 is observed in the bloodstream (*Figure 4—figure supplement 1*), mice were sacrificed, and the small intestine contents were collected (*Figure 4A*).

To assess how the gavage impacted the microbiome of the jejunum, where the bulk of fat absorption occurs (*Alfin-Slater and Aftergood, 2012*; *Borgstrom et al., 1962*), we sequenced the V4 region of 16S rRNA genes derived from DNA obtained from propidium monoazide (PMA) treated and untreated aliquots of each jejunal luminal sample. The PMA or 'live-only' aliquot, is depleted of DNA from cells with compromised membranes. In addition to live cells, the untreated or 'total' aliquot includes DNA from live as well as cells permeabilized by 18:2 and dead cells. This approach allowed us to gauge which taxa were still alive after the 18:2 treatment.

The effect of 18:2 on the microbial community was evident from analysis of the live cells but not for the total cell population: microbiomes within a diet-group could be distinguished by gavage treatment only when the live-only aliquot was analyzed (live-only; weighted UniFrac, n = 23 for LF diet: adonis, pseudo-F = 4.78, 15% of variance explained, p = 0.022; n = 21 for HF diet: adonis, pseudo-F = 7.84; 28% of variance explained, p = 0.003; also see *Figure 4—figure supplement 2A and B*). This observation suggests that 18:2 compromised select microbes, thereby decreasing their abundances and altering the abundances of other live microbes. Such differences in microbial abundances due to the 18:2 gavage should be evident by directly comparing the total and live-only aliquots for each sample. Indeed, we observed that for both diets, although the jejunal contents from saline-gavaged mice showed differences in the live-only and total diversity, this difference was greater in mice gavaged with 18:2 (p < 0.01 for LF diet, p < $10^{-7}$ for HF diet, Kruskal-Wallis tests) (*Figure 4B*). Hence, while compromised cells exist in the saline control animals, 18:2 caused additional cells to be compromised. We note that this difference (beta-diversity distance) was greater for the HF than the LF diet samples (*Figure 4B*), suggesting that the HF-diet conditioned microbiome was disrupted to a greater extent by 18:2 than the LF-diet microbiome.

The LF versus HF SBO diets themselves, on the other hand, had little effect on the microbiome. While the mice on the HF diet gained significantly more fat mass (p = 1.53*$10^{-4}$, mean in HF diet group = 0.043, mean in LF diet group = 0.028, 95% CI = (0.0076, 0.0218), two-sample, two-tailed t-test on epididymal fat pad mass), we observed no differences between the total microbiome composition of the jejuna of mice on the two diets (PERMANOVA on the total cell population, p > 0.5). Using a Kruskal-Wallis test, with FDR < 0.1, we observed that OTU 363731 mapping to *Akkermansia muciniphila* was 60-fold enriched in the HF diet. These results imply that the level of SBO and compensatory reduction in carbohydrates in the HF diet was not sufficient to greatly alter the microbiome.

OTUs 692154 and 592160, taxonomically assigned by Greengenes to *L. reuteri* and *L. johnsonii*, respectively, were the two most abundant lactobacilli OTUs in all samples. These OTUs displayed comparable relative abundances in the two diets (total aliquot; Kruskal-Wallis test and ANOVA on a linear mixed model to include cage effects, p values > 0.05, *Figure 4C and D*). These *L. reuteri* and *L. johnsonii* OTUs were present in the 18:2, live-only microbiota in both sets of mice (*Figure 4C and D*), suggesting these taxa survived the 18:2 acute treatment regardless of the dietary fat content. Note we detected *L. reuteri* OTU 692154 at very low levels in the microbiota of mice housed in three out of six LF diet cages and in two out of six HF diet cages (*Figure 4—figure supplement 2C*). Comparison of the relative abundance of these two OTUs in the total and live-only microbiota revealed these lactobacilli (with the exception of *L. reuteri* in the LF diet) enriched 2- to 5-fold (ANOVA on a linear mixed model to include cage effects and Kruskal-Wallis tests, p values < 0.01) after 18:2 gavage. Furthermore, the live-only microbiota of HF diet mice had an enrichment of 11 lactobacilli OTUs after 18:2 gavage (5- to 9-fold enrichment compared to control gavage, Kruskal-Wallis, FDRs < 0.1, *Figure 4—figure supplement 2D*) at the expense of *Allobaculum spp.* Similar

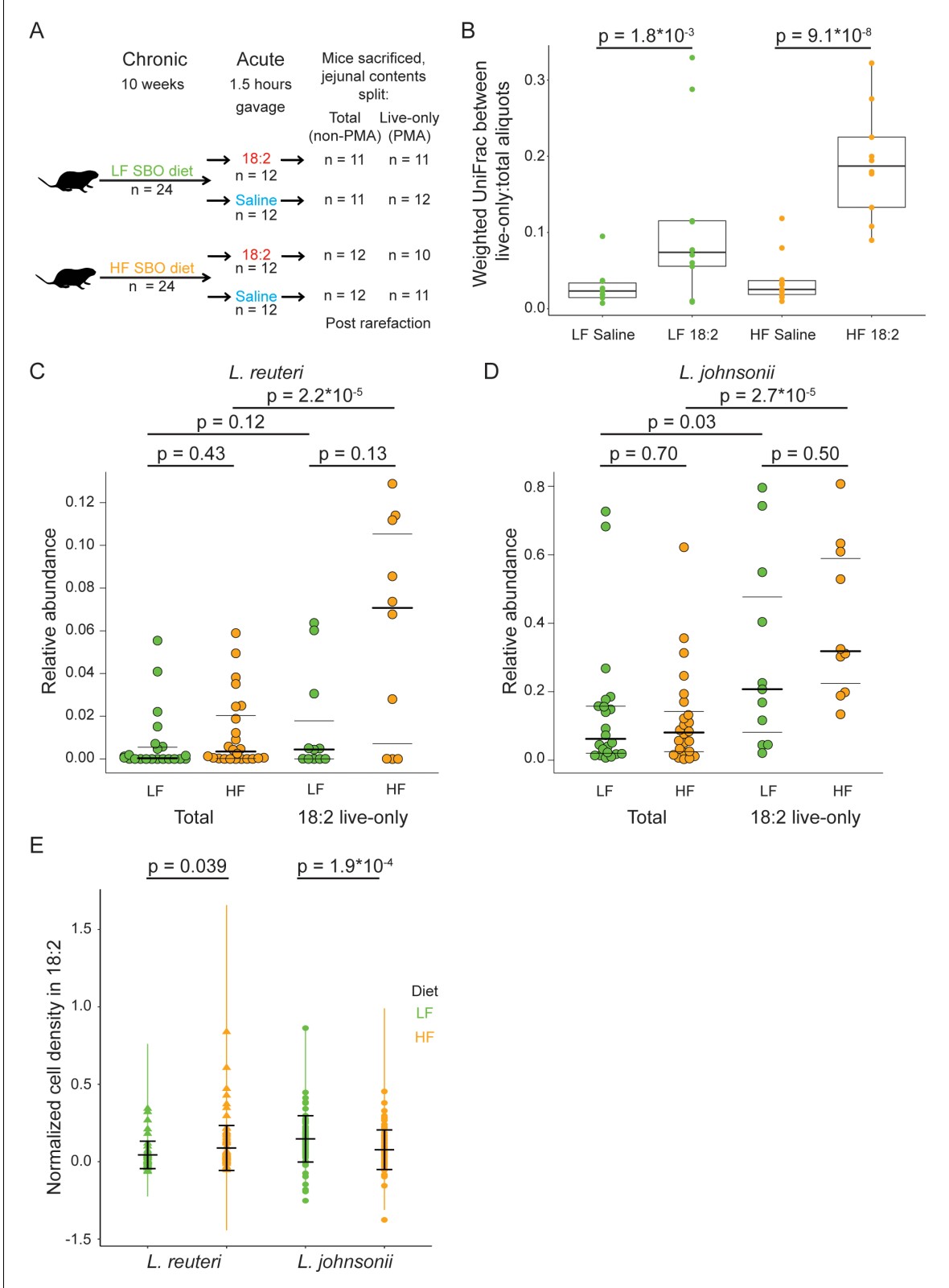

**Figure 4.** *L. reuteri* and *L. johnsonii* can survive 18:2 in vivo without 18:2 resistance. (**A**) Schematic of the SBO diet experiment. After 10 weeks on either the LF or HF SBO diet, 24 mice on each diet were gavaged with 18:2 or saline (n 12 for each). 1.5 hr post gavage, mice were sacrificed, jejunal contents collected and split into two. One aliquot was PMA treated (live-only cells) and the other was not (live and dead cells, total). 16S rRNA gene sequencing was performed on both aliquots. Sample size values shown on the right of the figure panel reflect samples passing rarefaction. (**B**) Weighted UniFrac

*Figure 4 continued on next page*

Figure 4 continued

distances between the live-only and total aliquots for each mouse sample. Significance values were determined using Kruskal-Wallis tests. For LF samples, mean in saline gavage group = 0.03, mean in 18:2 gavage group = 0.11; for HF samples, mean in saline gavage group = 0.04, mean in 18:2 gavage group = 0.19. (C) Relative abundance of *L. reuteri* OTU 692154 and D) *L. johnsonii* OTU 592160 in the total cell and 18:2 gavage live-only aliquots. For the total cell aliquots, post rarefaction, n = 22 for LF diet; n = 24 for HF diet. For the 18:2 gavage live-only aliquots, post rarefaction, n = 11 for LF diet; n = 10 for HF diet. Dark lines indicate the 50% quartile, and the two thinner lines show the 25% and 75% quartiles. Mean values for *L. reuteri*: total-LF diet = 0.007; total-HF diet = 0.012; 18:2-live-only-LF = 0.015; 18:2-live-only-HF = 0.061. Mean values for *L. johnsonii*: total-LF diet = 0.140; total-HF diet = 0.121; 18:2-live-only-LF = 0.302; 18:2-live-only-HF = 0.404. Significance values were determined using an ANOVA on a linear mixed model to include cage effects. For comparisons within diets, between total and live-only aliquots, similar results were obtained if only 18:2 gavaged animals were considered in the total aliquot. As well, similar results were obtained using Kruskal-Wallis tests. (E) Normalized cell density in 18:2 of lactobacilli isolated from mice on the low or high SBO diets. A value of one means cells were not inhibited by 18:2. For reference, the evolved strain LJ41072, which has enhanced resistance to 18:2 (*Figure 3D*), gives a value of 0.6. Black lines indicate the mean and standard deviations of the entire set of colonies. The colored lines show the standard deviations for replicate tested colonies. For *L. reuteri*, we excluded 7/120 isolates that failed to grow in medium lacking 18:2 and tested 113 isolates derived from 15 mice housed in 8 of 12 cages (5 HF diet cages and 3 LF diet cages; note that *L. reuteri* was not detected by 16S rRNA gene amplicon sequencing in several of the cages). For *L. johnsonii*, we excluded 33/192 isolates that failed to grow in medium lacking 18:2 and tested 159 isolates from 22 mice in all 12 cages. A single *L. reuteri* replicate gave a normalized cell density in 18:2 above 1.0. Significance values were determined by Kruskal-Wallis tests: for *L. reuteri* p = 0.039, mean in LF group = 0.04, mean in HF group = 0.09; for *L. johnsonii* p = $1.9*10^{-4}$, mean in LF group = 0.15, mean in HF group = 0.08. See also *Figure 4—figure supplement 1–4* and *Figure 4—source data 1*.

DOI: https://doi.org/10.7554/eLife.32581.013

The following source data and figure supplements are available for figure 4:

**Source data 1.** SBO mouse diets.
DOI: https://doi.org/10.7554/eLife.32581.019
**Figure supplement 1.**
DOI: https://doi.org/10.7554/eLife.32581.014
**Figure supplement 2.** Lactobacilli survive acute and chronic 18:2 exposure *in murine*.
DOI: https://doi.org/10.7554/eLife.32581.015
**Figure supplement 3.** Lactobacilli survive chronic 18:2 exposure *in murine*.
DOI: https://doi.org/10.7554/eLife.32581.016
**Figure supplement 4.** The 18:2 resistance of lactobacilli isolates in vitro does not relate to ability to survive acute 18:2 exposure *in murine*.
DOI: https://doi.org/10.7554/eLife.32581.017
**Figure supplement 5.** An HF diet isolated *L. reuteri* is resistant to 18:2.
DOI: https://doi.org/10.7554/eLife.32581.018

enrichment of live lactobacilli after the 18:2 gavage was observed for the LF diet, although no OTU passed our significance threshold. These observations suggest that lactobacilli resist acute 18:2 exposure particularly in the context of a high-18:2 diet.

To confirm that the *Lactobacillus* population was not reduced by the 18:2 gavage and that any changes in their relative abundances were due to die-offs of other bacteria, we quantified their levels in total and live cell fractions by qPCR. We determined the difference in the copy number of *Lactobacillus* 16S rRNA sequences in the total and live-only samples normalized to the equivalent difference for total Eubacteria. We observed no difference between the saline and 18:2 gavage samples for either diet (*Figure 4—figure supplement 2E*, two-sample, two-tailed t-test, p values > 0.1). All live-only to total relative copy numbers were close to 1, as expected if the *Lactobacillus* population was not reduced by 18:2 exposure.

To determine if our findings were limited to our specific mouse experiment, we repeated the chronic 18:2 exposure with two additional sets of mice originating from Taconic Biosciences and an F2 generation of mice from Jackson Laboratories. In these two additional sets of mice, 16S rRNA gene sequence diversity analysis of jejunal contents showed that the same two OTUs annotated as *L. reuteri* and *L. johnsonii* were again the predominant lactobacilli, although these are extremely unlikely to be the same lactobacilli strains present in our first study. In Taconic mice, *L. reuteri* and *L. johnsonii* were detected in the jejunum after 10 weeks on both diets (*Figure 4—figure supplement 3A and B*). In F2 Jackson mice, *L. johnsonii* was detected after 10 weeks on both diets (*Figure 4—figure supplement 3C*), whereas *L. reuteri* was only present in LF diet mice (*Figure 4—figure supplement 3D*). *L. reuteri*, however, was not observed in fecal samples from week 0 (*Figure 4—figure supplement 3E*). As all mice were similarly handled, the diets sterilized, and the mice bred in the

same facility, *L. reuteri* may have invaded the LF mice, though we cannot rule out the possibility of *L. reuteri* existing below detection. Nevertheless, these additional studies support the notion that lactobacilli populations are minimally impacted by chronic dosing of 18:2.

### *L. reuteri* isolated from SBO diet mice are sensitive to 18:2, but HF diet isolates show increased 18:2 resistance

Our results in mice suggest that *L. reuteri* and *L. johnsonii* survived chronic and acute exposure to 18:2 either directly, by 18:2 resistance, or indirectly, through an unknown aspect of life within the mouse gut. To assess the direct resistance of these lactobacilli to 18:2, we established a collection of *L. reuteri* and *L. johnsonii* isolates derived from the upper ileum (as a proxy for the jejunum) of mice on both HF and LF diets. We determined the ability of these isolates to grow in liquid culture amended with 18:2. While most isolates were sensitive to 18:2, we observed that *L. reuteri* isolates recovered from the HF-diet fed mice were on average more resistant to 18:2 than *L. reuteri* isolated from the LF-diet fed mice (113 isolates from 15 mice in eight cages Kruskal-Wallis, p < 0.05, *Figure 4E* and *Figure 4—figure supplement 4A*). This observation is consistent with the hypothesis that chronic exposure to a diet high in 18:2 promotes resistance in the resident *L. reuteri* population.

### Lactobacilli population-level 18:2 resistance in vivo does not predict the resistance of isolates in vitro

Next, we sought to relate the in vitro resistance of the *L. reuteri* isolates to the in vivo changes in *L. reuteri* populations before and after acute 18:2 exposure. To do so, we assessed the enrichment of *L. reuteri* OTU 692154 in the live jejunal aliquot post 18:2 gavage: we considered the rarified sequence counts for this OTU in the live-only aliquot (i.e., in PMA-treated samples) in mice gavaged with 18:2 normalized by the equivalent sequence counts for the OTU in saline gavaged co-caged mice. A resulting $\log_{10}$ ratio greater than 0 indicates that live *L. reuteri* OTU 692154 had greater relative abundance counts in mice gavaged with 18:2 compared to same-cage controls gavaged with saline, signifying that other OTUs had been depleted. We observed no correlation between the ability of these strains to grow in vitro in 18:2 and their abundance in mice gavaged with 18:2 for mice on either diet (*Figure 4—figure supplement 4B and C*). Note that we cannot exclude the possibility that the isolation procedure favored susceptible strains, and thus is not representative of the in vivo population. With this caveat in mind, these results indicate that while chronic exposure to 18:2 can result in *L. reuteri* strains with higher 18:2 resistance, the mouse gut environment protects susceptible strains.

We partially replicated these findings with *L. johnsonii*: all isolates were sensitive to 18:2, but *L. johnsonii* from the HF diet-fed mice were more strongly inhibited by 18:2 than those isolated from the LF-fed mice (159 isolates from 22 mice in 12 cages; Kruskal-Wallis, p < 0.001, *Figure 4E*). Therefore, the results for *L. johnsonii* are similar to those of *L. reuteri*, with a lack of congruence between the response of the population in vivo and the resistance of isolates in vitro.

### Putative fatty acid responsive genes are mutated in HF diet isolated *L. reuteri*

We sequenced to 50X coverage an isolate of *L. reuteri* resistant to 18:2, derived from a HF diet mouse (strain LRHF, *Supplementary file 2*, *Figure 4—figure supplement 5*). Although we cannot be certain that HF diet-isolated *L. reuteri* share a common ancestor with those present in LF diet mice, we compared LRHF to LR0, the 18:2-susceptible isolate from a LF diet mouse and used in the in vitro evolution assay. The comparison revealed 71 mutations in 60 genes with functions predominantly in DNA metabolism, energy metabolism, and environmental response (*Table 3*, *Supplementary file 3*). None of the genes mutated in the in vitro evolution assay differed between LR0 and LFHF. LRHF exhibited mutations in a sodium-hydrogen antiporter gene and a peroxide stress (*PerF*) gene, both of which may represent adaptation to an acidic environment caused by exposure to FAs. Of potential relevance to FA exposure, we observed mutations in a membrane-bound lytic murein transglycosylase D precursor involved in the production of the peptidoglycan layer (*Vollmer et al., 2008*) and the fructosyltransferase *Ftf* involved in the production of exopolysaccharide (*Sims et al., 2011*). These results suggest that exposure to 18:2 in vivo does not invoke selection on the same genes that are implicated in 18:2 resistance in vitro.

**Table 3.** Nonsynonymous mutations in genes with known function in LRHF.

| Gene | Functional group | Mutation type |
|---|---|---|
| Helicase | DNA metabolism | NS |
| N-acetyl-L,L-diaminopimelate aminotransferase | energy metabolism | NS |
| Mrr restriction system protein | DNA metabolism | NS |
| Putative NADPH-quinone reductase | energy metabolism | NS |
| Accessory gene regulator C (sensor histidine kinase) | environmental response | NS |
| Transcriptional regulator, XRE family | environmental response | FS |
| ATPase component BioM of energizing module of biotin ECF transporter | energy metabolism | NS |
| CRISPR-associated protein, Csn1 family | DNA metabolism | NS |
| Transcriptional regulator, XRE family | environmental response | NS |
| Exodeoxyribonuclease VII small subunit | DNA metabolism | NS |
| Type I restriction-modification system, specificity subunit S | DNA metabolism | NS |
| ABC1 family protein | energy metabolism | NS |
| Protein serine/threonine phosphatase PrpC, regulation of stationary phase | energy metabolism | FS |
| Nucleotide sugar synthetase-like protein | DNA metabolism | NS |
| DNA repair protein RecN | DNA metabolism | NS |
| ABC transporter substrate-binding protein | energy metabolism | FS |
| Fructosyltransferase Ftf | membrane | PS |
| Oxidoreductase | energy metabolism | NS |
| Ribonuclease M5 | DNA metabolism | NS |
| Zinc-containing alcohol dehydrogenase; quinone oxidoreductase | energy metabolism | FS |
| DinG family ATP-dependent helicase YoaA | DNA metabolism | NS |
| Aromatic amino acid aminotransferase gamma | energy metabolism | NS |

Mutations: NS = nonsynonymous; FS = frameshift; PS = premature stop. See also *Figure 4—figure supplement 5* and *Supplementary file 3* and *5*.

DOI: https://doi.org/10.7554/eLife.32581.020

# Discussion

A drastic change in dietary macronutrient composition has the capacity to restructure the microbiome within a day (*David et al., 2014b*; *Faith et al., 2011*; *Turnbaugh et al., 2009*) and is one of the most influential contributors to microbiome composition (*Carmody et al., 2015*). Here, we consider how the gut microbiome is influenced by diet from the perspective of a single FA known to be toxic to gut microbes: specifically, the interaction between lactobacilli and linoleic acid (18:2). In accord with previous reports, we observed 18:2 to inhibit the growth of most naturally-derived lactobacilli in vitro. However, in the mouse gut, *L. reuteri* and *L. johnsonii* persisted through both chronic and acute exposures to 18:2. *L. reuteri* isolates derived from mice on a diet high in 18:2 included some that were more resistant to 18:2. This observation suggests that 18:2 resistance has the potential to be selected in a host. In vitro, *L. reuteri* and *L. johnsonii* both evolved 18:2 resistance through mutations in the cell wall/membrane and fat metabolism genes. Collectively, these data indicate that the host gut environment protects gut microbes from the inhibitory effects of FAs, but that these microbes can also evolve resistance, providing additional resilience.

The mutations our 18:2 in vitro adapted lactobacilli strains acquired are consistent with the known bacteriostatic and bactericidal mechanisms of 18:2: by increasing membrane fluidity and permeability (*Greenway and Dyke, 1979*) potentially leading to cell lysis or leakage (*Galbraith and Miller, 1973b*; *Parsons et al., 2012*), by blocking absorption of essential nutrients (*Nieman, 1954*), and by inhibiting FA synthesis (*Zheng et al., 2005*) and oxidative phosphorylation (*Galbraith and Miller, 1973a*). Lactobacilli are also capable of combating 18:2 toxicity by converting 18:2 to conjugated 18:2 and subsequently a monounsaturated or saturated fatty acid (*Jenkins and Courtney, 2003*; *Kishino et al., 2013*). We did not recover any mutations in genes known to be involved in the production of conjugated 18:2.

Despite the toxicity of 18:2 towards lactobacilli, mouse-associated *L. reuteri* and *L. johnsonii* were present at equivalent relative abundances in mice fed diets high or low in 18:2. Moreover, these microbes survived a gavage of 18:2 equal to double what mice normally encounter in their daily diet. Our results are consistent with the findings of Holmes and colleagues, who analyzed the fecal microbiomes of mice on 25 different SBO diets varying in their macronutrient (fat, protein, carbohydrate) composition. Their results demonstrate that fat has only a minor effect on microbiome structure (*Holmes et al., 2017*). In contrast, in microbial systems engineered for waste processing, concentrations of linoleic acid within the range predicted to be consumed by animals can cause failure of the desired microbial biodegradation processes (*Lalman and Bagley, 2000*). The resistance of lactobacilli to linoleic acid in the mouse host is therefore inferred to be dependent on the complexity of the gut habitat.

In mice, lactobacilli colonize both the small intestine and forestomach (*Walter et al., 2007*). While lingual lipases exist in mice (*DeNigris et al., 1988*), fat digestion occurs primarily in the small intestine. As a result, forestomach microbes should not be exposed to a high concentration of free FAs, and SBO itself is not toxic. A gavage of 18:2, on the other hand, exposes forestomach microbes to free 18:2. Lactobacilli may be protected from this direct exposure by their capacity to form a dense biofilm on non-mucus secreting stratified epithelial cells (*Frese et al., 2013*). In the human host, other aspects of the small intestinal habitat likely buffer the microbiota.

The decline of *L. reuteri* in Western populations may never be fully explained. In the 1960's and 1970's prior to the emergence of SBO as a major dietary fat source, *L. reuteri* was recovered from the intestinal tract of 50% of subjects surveyed and was considered a dominant *Lactobacillus* species of the human gut (*Reuter, 2001*). Today, however, it is found in less than 10% of humans in the USA and Europe (*Molin et al., 1993*; *Qin et al., 2010*; *Walter et al., 2011*), yet it is present at a reported 100% prevalence in rural Papua New Guineans (*Martínez et al., 2015*). Moreover, human *L. reuteri* strains show very little genetic variation (*Duar et al., 2017*; *Oh et al., 2010*), and one human associated lineage of *L. reuteri* appears to have arisen approximately when SBO consumption increased (*Walter et al., 2011*). These observations raise the question of whether a change in dietary habits drove the decline in the prevalence of *L. reuteri* in Western populations. In humans, *L. reuteri* forms neither high gastric populations nor biofilms (*Frese et al., 2011*; *Walter, 2008*), thus human-derived *L. reuteri* strains may have survived increased exposure to 18:2 by developing resistance. Indeed, we did observe that some human *L. reuteri* strains are resistant to 18:2, but not all. While the increase in SBO consumption may have conspired with other facets of modernization to reduce the prevalence of *L. reuteri* in Western populations, it did not appear to have resulted in a selective sweep of 18:2 resistant *L. reuteri*.

The mechanistic underpinnings of how dietary components shape the composition of the gut microbiome need to be further elucidated if manipulation of the microbiome for therapeutic applications is to succeed. Dietary components have the potential to inhibit microbes directly through their toxicity, or indirectly by promoting the growth of other, more fit, microbes. While FAs are generally toxic to many lactobacilli, this work suggests that toxicity is greatly reduced when lactobacilli are host-associated. Future work in this area will elucidate how the host environment protects gut microbes from otherwise toxic dietary components such as FAs, and the ways specific strains within the microbiome can be resilient to such stresses.

## Materials and methods

### Strains

*Supplementary file 1* details the naturally derived *L. reuteri* strains from various hosts and countries. The *L. reuteri* strain (LR0) and *L. johnsonii* strain (LJ0) used in the in vitro 18:2 evolution assay were isolated from the jejunum contents of a mouse originally purchased from Taconic Biosciences (Hudson, NY, USA) and maintained on the low fat soybean oil diet for 6 weeks since weaning, and strain LRHF was isolated from a parallel mouse on the high fat soybean oil diet for 6 weeks since weaning (see Mouse care section for further details).

## Media and culturing

Lactobacilli were cultured in MRS liquid medium (Criterion, Hardy Diagnostics, Santa Maria, CA, USA) or on MRS agar plates (Difco, BD, Sparks, MD, USA), pH-adjusted to 5.55 using glacial acetic acid. All liquid cultures and plates were incubated at 37°C in an anoxic chamber (Coy Lab Products, Grass Lake, MI, USA) supplied a gas mix of 5% $H_2$, 20% $CO_2$, and 75% $N_2$.

## Disc diffusions

We plated 100 µl of a dense, overnight culture of *L. reuteri* strain ATCC 53608 on an agar plate and applied sterile Whatman paper (Buckinghamshire, UK) discs to the surface of the culture plate. To each disc, we added 10 µl of each test compound or control. Compounds tested were alpha-linolenic acid (18:3) ($\geq$99%, L2376, Sigma Aldrich, St. Louis, MO, USA), linoleic acid (18:2) ($\geq$99%, L1376, Sigma Aldrich), oleic acid (18:1) ($\geq$99%, O1008, Sigma Aldrich), stearic acid (18:0) ($\geq$98.5%, S4751, Sigma Aldrich), palmitic acid (16:0) ($\geq$99%, P0500, Sigma Aldrich), 0.85% NaCl (saline), DMSO, glycerol, all afore-mentioned FAs mixed (FA mix), the FA mix with glycerol, and soybean oil (Wegmans, NY, USA). FAs were dissolved in DMSO to a concentration of 50 mg/ml, except for stearic acid, which was dissolved to a concentration of 5 mg/mL due to its lower solubility. For the FA mix, the five FAs were mixed in the ratio that these FAs are present in soybean oil: 14% 16:0, 4% 18:0, 23% 18:1, 52% 18:2, 6% 18:3. For the FA mix with glycerol, glycerol was mixed with the FA mix to a molar mass ratio of 0.1 (e.g., the molar mass ratio of glycerol in the total molar mass of soybean oil). For testing glycerol alone, the same amount of glycerol used in the FA mix with glycerol was used, and the total volume was brought up to 10 µl with DMSO. Plates were dried for 20 min at 37°C before being turned agar side up and incubated overnight.

## Live/dead assay

First, we centrifuged 5 mL of an overnight culture of *L. reuteri* ATCC 53608 at 10,000 rcf for 10 min and resuspended the pellets in 30 mL of 0.85% NaCl solution. Then we centrifuged 1 mL aliquots of the resuspended culture at 15,000 rcf for 5 min. The resulting pellets were resuspended in 0.85% NaCl solution to a total volume of 1 mL in the presence of 18:2, 18:3, 0.85% NaCl, or ethanol. We diluted FAs in 100% ethanol in a ten-fold dilution series ranging from 0.01 to 1000 µg/ml. We incubated samples at room temperature for 90 min on a rocking platform (setting 6; VWR, Radnor, PA, USA) and inverted the samples by hand every 20 min to ensure adequate mixing. After exposure to the FA, we washed the cells by centrifuging at 15,000 rcf for 5 min, and resuspending the pellets in 1 mL 0.85% NaCl; we repeated this wash a second time. To measure the permeability of the cells, we stained samples using the Live/Dead BacLight Bacterial Viability Kit (L7007, Invitrogen, Life Technologies, Grand Island, NY, USA) according to the manufacturer's instructions. We measured fluorescence from propidium iodide and SYTO9 on a BioTek Synergy H1 Hybrid Reader (BioTek Instruments, Inc., VT, USA). At each FA concentration, fluorescence was read in triplicate (technical replicates). We used the drc package (*Ritz et al., 2015*) in R (*Team, 2016*) for dose-response modeling and statistical analyses.

## Estimated concentrations of linoleic acid in the mouse small intestine

The mice in this study consumed on average 2.7 grams of mouse food per day. Therefore, mice on a 23% by weight soybean oil mouse diet (the 44% by calorie HF diet), consume 0.62 grams SBO. In SBO, fatty acids comprise 90% of the molar mass. As 52% of the fatty acids in SBO are 18:2, therefore in a day, a mouse consumes ~0.3 grams 18:2. We estimated that the transit time of fat from feeding and into the bloodstream is approximately 1.5 hr (*Figure 4—figure supplement 1A*). Using the approximation that food is consumed continuously over the course of the day, we expect 18 µg of 18:2 to pass through the small intestine in a 1.5 hr period. The volume of the small intestine is between 200 and 500 µl (*McConnell et al., 2008*) and therefore approximately 36 to 91 µg/ml 18:2 will pass through the small intestine in a transit time. For the 7% by weight SBO diet (16% by calorie LF diet), 11 to 28 µg/ml 18:2 will pass in a transit time.

## Linoleic acid liquid growth assay

We inoculated a *Lactobacillus reuteri* or *L. johnsonii* colony grown 1 to 2 days on an MRS agar plate into a well containing 300 µl MRS liquid medium on a sterile 2 ml 96 well polypropylene plate

(PlateOne, USA Scientific, FL, USA). We covered the plate with Breathe-Easy polyurethane film (USA Scientific, FL, USA) and incubated the plate overnight at 37°C in an anoxic chamber (Coy Lab Products, Grass Lake, MI, USA) supplied a gas mix of 5% $H_2$, 20% $CO_2$, and 75% $N_2$. Following overnight growth, we split the cultures 100-fold into a new 96 well plate, whereby each overnight culture was diluted into a well containing MRS medium and to a well containing MRS medium plus 1 mg/ml linoleic acid. To emulsify the FA in solution, prior to and following inoculation, we vortexed the 2 ml plate on a Multi-Tube Vortexer (VWR, PA, USA) for 30 s at setting 3.5. We then transferred the entire plate to a 300 µl Microtest Flat Bottom non-tissue treated culture plate (Falcon, Corning, NY, USA). We measured the $OD_{600}$ of the plate on a BioTek Synergy H1 Hybrid Reader (BioTek Instruments, Inc., VT, USA) at approximately 0, 2, 4, 6, and 8 hr. For growth curves of strains LR0, LR2-1, LRHF, LJ0, and LJ41072, cultures were read in triplicate (technical replicates).

We quantified how well the strain grew in 18:2 compared to the without 18:2 control by analyzing the last three time points of the growth assay. We used this approach over fitting a doubling time because, in the first few points of the growth curve, the OD values in wells with cells and 18:2 were lower than those in inoculation-control wells (e.g., with 18:2, but lacking cells). Hence, for the first few time points when subtracting the $OD_{600}$ of medium with 18:2, without cells from the $OD_{600}$ of medium with 18:2, with cells, we obtained negative $OD_{600}$ values. As well, the time spent in log phase varied among the strains and proper modeling of log to late-log phase could not be achieved without significant trimming and manipulation of the data. At these final three time points, we determined the ratio of the 'blanked $OD_{600}$s' for the strain growing in MRS medium with linoleic acid to the strain growing in MRS medium alone:

$$\frac{OD600_{MRS\ with\ 18:2}}{OD600_{MRS}}$$

We excluded time points in which the $OD_{600}$ in MRS medium alone was less than 0.1 (i.e. strain did not grow). We determined the mean of the above ratios for the last three time points. All negative normalized cell densities were confirmed to result from negative values in the $OD_{600}$ of cells growing in 18:2.

For the naturally derived *L. reuteri* strains, we tested strains in triplicate to sextuplet (biological replicates) and averaged replicate normalized cell densities. For each *L. reuteri* and *L. johnsonii* strain isolated from mice on the SBO diet, we tested eight isolates from two mice per cage. The sample sizes for the SBO diet mice isolated strains were upper bounded by the observation that the microbiomes of these mice were dominated by one or few *L. reuteri/johnsonii* OTUs. Sixty-two isolates were tested between 2 and 5 times (biological replicates) and normalized cell densities were averaged across replicates. Statistical analyses were completed using kruskal.test in the R stats package (*Team, 2016*).

## In vitro evolution of 18:2 resistant lactobacilli

For *L. reuteri* strain LR0 and *L. johnsonii* strain LJ0, both originating from a mouse on the LF SBO diet for 6 weeks, we inoculated a single colony into 5 ml MRS and grew the cultures overnight. The following day, we diluted the overnight cultures for LR0 and LJ0 100-fold, separately, into five 5 ml MRS medium supplemented with 5 mg/ml 18:2. These five cultures became the five populations evolved for *L. reuteri* or *L. johnsonii* and we refer to them as LR1-5 and LJ1-5, respectively. We passaged these cultures twice daily using a 100-fold dilution. We omitted an emulsifier (DMSO or ethanol) from this assay to avoid the possibility of the lactobacilli adapting to the emulsifier rather than to 18:2. As a result, we needed to use a relatively high concentration of 18:2. To promote and maintain emulsification of the FA, we rigorously vortexed the tubes every few hours throughout the day. After seven days, we increased the concentration of 18:2 to 6 mg/ml. Each subsequent week, we increased the concentration by 1 mg/ml until reaching a final concentration of 10 mg/ml. Each week, we froze a 20% glycerol stock of each population at −80°C. We excluded *L. johnsonii* population #1, LJ1, from further study due to contamination.

## Whole genome sequencing of *Lactobacillus* populations and isolates

We isolated genomic DNA from approximately 30 µl cell pellets frozen at −20°C using the Gentra Puregene Yeast/Bact. Kit (Qiagen, MD, USA). For isolates, we grew a single 50 ml log to late-log

phase culture from a single colony. For populations, we inoculated five 10 ml cultures directly from glycerol stock, grew the cultures to log to late-log phase, and thoroughly mixed the replicate cultures together before pelleting to aid in representing the diversity of the original population structure. We grew 18:2-adapted isolates and populations in MRS medium with 10 mg/ml 18:2, and nonadapted isolates in MRS medium. We used the Gentra Puregene Yeast/Bact. kit following the optional protocol adjustments: a 5 min incubation at 80°C following addition of the Cell Lysis Solution, a 45 min to 60 min incubation at 37°C following RNase A Solution addition, and a 60 min incubation on ice following addition of Protein Precipitation Solution. DNA was resuspended in Tris-EDTA and further purified using the Genomic DNA Clean and Concentrator−25 (Zymo Research, CA, USA). We quantified isolated DNA using the Quant-iT PicoGreen dsDNA Assay Kit (Thermo Fisher Scientific MA, USA). Lastly, to ensure we had obtained large molecular weight DNA, we ran the DNA on a 1% sodium borate agarose gel (Agarose I, Amresco, OH, USA).

We prepared barcoded, 350 bp insert libraries using the TruSeq DNA PCR-Free Library Preparation Kit (Illumina, CA, USA). We fragmented starting genomic DNA (1.4 µg) using the recommended settings on a Covaris model S2 (Covaris, MA, USA). The barcodes used for each library are indicated in *Supplementary file 2*. We submitted these barcoded libraries to the Cornell University Institute of Biotechnology Resource Center Genomics Facility where they were quantified by digital PCR using a QX100 Droplet Reader (Bio-Rad Laboratories, CA, USA), pooled (*Supplementary file 2*), and pair-end sequenced on an Illumina MiSeq 2 × 300 bp platform using reagent kit V3 (Illumina, CA, USA). Resulting reads from libraries sequenced on multiple MiSeq runs were merged for further analyses.

## Genome assembly of *Lactobacillus* populations and isolates

To generate reference genomes for the ancestor strains used in the in vitro evolution assay, we assembled paired-end sequences for *L. reuteri* LR0 and *L. johnsonii* LJ0 using SPAdes v3.7.1 (*Nurk et al., 2013*; *Prjibelski et al., 2014*) with k-mers 21, 33, 55, 77, 99, and 127 using the 'careful' option to reduce mismatches and indels. To select and order contigs, we aligned the assembled genomes against the closest complete genome available: NCC 533 for *L. johnsonii* and TD1 for *L. reuteri* as determined by a whole genome alignment using nucmer in MUMmer (*Kurtz et al., 2004*). The assembled genomes we aligned against the NCC 533 or TD1 genome using ABACAS.1.3.1 (*Assefa et al., 2009*) with the 'nucmer' program. Next, we aligned previously unaligned contigs using promer. We merged these sets of aligned contigs into one file and contigs with low coverage, less than 20, were removed. Finally, we ordered these filtered contigs using promer without the maxmatch option (-d) to prevent multiple reference-subject hits. For the LR0 genome, we identified a contig representing a plasmid from the assembly and included it in the set of assembled contigs. We uploaded these assembled genomes to RAST (*Aziz et al., 2008*; *Brettin et al., 2015*; *Overbeek et al., 2014*) for annotation (see *Supplementary file 2* for details on the assembled genomes).

## Variant allele detection in 18:2 resistant lactobacilli

First, we manually identified variant alleles in an isolate from *L. reuteri* population LR2, LR2-1, and an isolate from *L. johnsonii* population LJ4, LJ41072, using the Integrative Genomics Viewer (*Robinson et al., 2011*; *Thorvaldsdóttir et al., 2013*). We used the variants in these isolates to calibrate the allele detection methods applied to the whole populations. Next, we identified variant alleles in the populations by aligning the paired-end sequence reads to the ancestor genome (LR0 or LJ0) using BWA-MEM (*Li and Durbin, 2009*). We marked duplicate sequences using Picard 2.1.1 (http://broadinstitute.github.io/picard) and utilized Genome Analysis Toolkit (GATK) (*McKenna et al., 2010*), and the GATK Best Practices recommendations (*DePristo et al., 2011*; *Van der Auwera et al., 2013*) to accurately select true variants. This pipeline realigns indels and recalibrates and filters base calls using the known alleles identified in the isolates using a BQSR BAQ gap open penalty of 30. We used the GATK HaplotypeCaller to call alleles with the maxReadsInRegionPerSample option set utilizing the observed coverage binned across the genome by the GATK DepthOfCoverage script. We applied the following options for populations and isolates: pcr_indel_model was set to 'NONE', stand_call_conf was set at '10', stand_emit_conf at '4'. For populations only, we set sample_ploidy at '10' and for isolates, '1'. After we had separately processed all

populations and isolates, we jointly called alleles across the entire set of populations and isolates using GenotypeGVCFs with sample_ploidy at '10', stand_call_conf at '10', and stand_emit_conf at '4'.

We filtered these results to remove alleles with frequencies less than 10% and to remove alleles in genes annotated with 'mobile element protein', 'transposase', 'phage', or 'RNA'. In addition, the ancestor genomic reads were mapped onto the ancestor genome to aid in the removal of poorly mapping reads. We removed alleles discovered in the evolved isolates and populations that were also present at frequencies greater than 0.5 in the aligned ancestor reads against the reference. The remaining alleles we manually checked using IGV to remove any alleles in regions of the genome with abnormally high coverage, compared to the directly adjacent regions, likely representing genomic repeat regions. Filtered and unfiltered reads are presented in *Supplementary file 3* and *4*.

## Analysis of mutated genes

We used PredictProtein (*Yachdav et al., 2014*) to predict the cellular location and structure of hypothetical and putative proteins and SignalP 4.0 (*Petersen et al., 2011*) to predict signal peptide sequences.

## Generation of *L. reuteri* mutants

To test the role of the mutations discovered in the in vitro evolution experiment on fatty acid resistance, we recreated the *L. reuteri* mutations in the recombineering strain PTA 6475 using the procedure described by *van Pijkeren and Britton (2012)*. Briefly, *L. reuteri* ATCC PTA 6475 (BioGaia AB, Sweden) bearing the plasmid pJP042, which has inducible RecT and is selectable with 5 µg/ml erythromycin, was induced with 10 ng/ml peptide pheromone (SppIP) (Peptide 2.0, VA, USA) at $OD_{600}$ 0.55–0.65. After washing the cells in 0.5 M sucrose, 10% glycerol, we electroporated the cells with 100 µg of the recombineering oligo targeting the FabT or hydrolase gene and 40 µg of oligo oJP577 (*van Pijkeren and Britton, 2012*), which targets *rpoB*, rendering the cells rifampicin-resistant. We electroporated in 0.2 cm Gene Pulser cuvettes (Bio-Rad, CA, USA) using a Bio-Rad Gene Pulser Xcell with conditions 2.5 kV, 25 µF, and 400 Omega. We recovered cells for 2 hr at 37°C and then plated the cells on MRS supplemented with 25 µg/ml rifampicin and 5 µg/ml erythromycin.

We screened resulting colonies using either a restriction digest or primers specific to the mutation through mismatch amplification mutation analysis-PCR (MAMA-PCR) (*Figure 3—source data 1*). For screening by restriction digest, we first amplified the FabT or hydrolase gene by colony PCR in 8 µl reactions: a small amount of a colony, 100 nM f.c. of each primer (see *Figure 3—source data 1*), and 1x Choice Taq Mastermix (Denville Scientific, MA, USA). PCR conditions were 94°C for 10 min, 35 cycles of 94°C for 45 s, 56 or 58.5°C (see *Figure 3—source data 1*) for 1 min, and 72°C for 30 s, followed by a final extension at 72°C for 10 min. Reactions were held at 10°C and stored at 4°C. Following, the PCR products were digested in 16 µl reactions at 37°C for 1 hr: 8 µl PCR product, 0.2 µl (four units) MfeI (NEB, MA, USA), and 1x CutSmart Buffer (NEB). For screening by MAMA-PCR, PCRs were carried out as before except an additional primer specific to the mutation was included. We confirmed that the mutations were correct by Sanger sequencing (GENEWIZ, NJ, USA) the entire FabT or hydrolase gene using PCR conditions and primers previously described. The pJP042 plasmid was lost from cells by passaging in MRS.

## Mouse experiments

All animal experimental procedures were reviewed and approved by the Institutional Animal Care and Usage Committee of Cornell University protocol 2010–0065.

## Mouse soybean oil diets

The 16% and 44% SBO diets were custom designed by and purchased pelleted, irradiated, and vacuum packed from Envigo (formerly Harlan Laboratories, Inc., Madison, WI, USA, www.envigo.com). We stored open, in-use diet bags at 4°C and unopened, bags at −20°C. See *Figure 4—source data 1* for the diet compositions. The increase of SBO in the HF diet was compensated by a decrease in cornstarch (carbohydrate). Also, the amounts of protein (casein), vitamins, and minerals were increased in the HF diet to prevent nutritional deficiencies from arising: HF diet fed mice consume a more calorically dense diet and thus intake a smaller volume of food per body mass.

## Determination of fatty acid transit time to the bloodstream

We gavaged nine mice with 6 mg per gram mouse weight 18:2. Every half hour following gavage, we euthanized a mouse by $CO_2$ asphyxiation and collected blood by cardiac puncture. Blood was collected into EDTA coated tubes and stored on ice. Tubes were spun at 900 rcf at 4°C for 10 min and plasma was collected and stored at −80°C. We extracted lipids using the Bligh and Dyer method (*Bligh and Dyer, 1959*) and quantified FA methyl esters on a Hewlett-Packard 5890 series II gas chromatograph with a flame ionization detector (GC-FID) using $H_2$ as the carrier. See *Su et al., 1999* for further details.

We used a linear mixed model to determine if the gavage treatments significantly altered the plasma levels of 18:2 and 18:3 in 1.5 hr. The model was *fatty acid mass ~diet + gavage + total fatty acid mass + (1|cage) + (1|GC run date) + (1|fatty acid extraction date) + plasma vol + (1|study)*, where the terms *cage, GC run date, fatty acid extraction date,* and *study* were handled as random effects and all others as fixed effects. *GC run date* refers to when the extracted fatty acids were run on the gas chromatograph, and *plasma volume* refers to the amount of mouse plasma used in the extraction. Models were run in R (*Team, 2016*) using the lme4 package (*Bates et al., 2015*) with REML = FALSE and the control optimizer set to 'bobyqa'. Significance values were determined using a two-sample, two-tailed t-test (t.test in the R stats package (*Team, 2016*) on the least squares means estimates data from the predict R stats function run on the model.

## Mouse care

In this study, we used three sets of male C57BL/6 mice bred in three different facilities: Jackson Laboratories (Bar Harbor, ME, USA), Taconic (Hudson, NY, USA), and an F2 generation of mice originally purchased from Jackson Laboratories. At weaning (3 weeks of age), we split littermates into cages housing up to four mice and provided the mice either the LF (16% kcal SBO) or HF (44% kcal SBO) diet (*Figure 4—source data 1*). Littermates were split so to balance mouse weights within a cage and between the two diets. All mice were housed in the Accepted Pathogen Facility for Mice at Cornell University.

In total, 24 mice were purchased directly from Jackson Laboratories and maintained in six cages on the LF diet and 24 mice in six cages on the HF diet; from Taconic, 12 mice in three cages on the LF diet and 12 mice in three cages on the HF diet; and the F2 mice from Jackson Laboratories were comprised of 11 mice in five cages on the LF diet and 15 mice in five cages on the HF diet. Sample sizes of five mice per group have been successful in delimiting diet-driven microbiome composition differences (*Turnbaugh et al., 2008*). The three different sets of mice were maintained at distinct time periods with the goal of ensuring our findings were not specific to a given base-microbiota. Up to four mice were co-caged. We stocked cages with Pure-o-cel (The Andersons, Maumee, Ohio, USA), cotton nestlets, and plastic igloos so to avoid the introduction of exogenous fat. Food was placed in the cages and not on the wire racks to minimize loss and crumb buildup of the diets as the HF SBO diet does not maintain pelleted form. Twice weekly, we completely replaced cages and food. We weighed the amount of new food provided. To obtain mouse weights, we weighed mice in plastic beakers at the same approximate time of day twice weekly. We collected fresh fecal samples once weekly from the beakers into tubes on dry ice, which were later stored at −80°C. Mice were handled exclusively inside of a biosafety cabinet. We changed personal protective equipment and wiped all surfaces with a sterilant between cages to prevent cross-contamination. To measure food consumption, we filtered food crumbs out of the used bedding using a large hole colander followed by a fine mesh sieve, weighed the recovered food, and subtracted this amount from the known amount of food provided.

After 10 weeks on the SBO diets, we gavaged the Jackson Laboratory mice with saline (0.85% NaCl) or 18:2. The Taconic mice were gavaged with phosphate buffered saline (PBS) or 18:2, and the F2 mice from Jackson Laboratories with PBS, 18:2, or 18:3. The volume gavaged was 6 mg per gram mouse weight. The amount of FA gavaged is roughly double the amount of 18:2 consumed by mice on the LF diet each day, and more than half of the 18:2 consumed per day by mice on the HF diet. Within a cage, we gavaged half of the mice with a FA and the other half with saline/PBS, selecting which mouse received which gavage so to balance mouse weights between gavage groups. Following gavage, we moved mice to a fresh cage supplied with water, but lacking food. After 1.5 hr, we euthanized mice by decapitation and harvested small intestine contents (see below).

## Processing of small intestine contents

To harvest the jejunal contents, we divided mouse small intestines into three equivalent pieces. For Jackson Laboratory mice, we flushed the middle segment, the jejunum, with 10 ml anoxic 0.85% NaCl using a blunt, 18G, 1.5 inch needle into a 15 ml conical tube that we immediately placed on ice. After flushing, we quickly shook the tube and split its contents roughly equally into a second 15 ml conical tube. One of the tubes we covered with foil to which we added 12.5 µl of propidium monoazide (PMA) (Biotium, Fremont, CA, USA; f.c. 50 µM from a 2 mM stock dissolved in DMSO). Which tube received PMA, the original or the second, we alternated between mice. To the other tube, we added 12.5 µl DMSO. To allow the PMA time to enter permeabilized cells, we placed all tubes on ice on a rocking platform for 5 min. To activate the azido group in PMA and cause DNA damage, we removed the foil from the tubes, placed the tubes horizontally on ice, and exposed the tubes for 5 min to a 650W halogen bulb (Osram 64553 C318, Danvers, MA, USA) positioned 20 cm from the samples. We frequently rotated the tubes during these 5 min to ensure equal light exposure across the whole sample. We immediately spun these tubes at 4500 rcf for 5 min at 4°C. After we discarded the supernatant, we flash froze the tubes on liquid $N_2$, placed them on dry ice, and later stored the tubes at −80°C. We also flushed the upper half of the last segment of the small intestine, the ileum, with MRS medium and 20% glycerol, immediately placed the glycerol stock on dry ice, which we later stored at −80°C. For the other mice, we flushed the jejunum with 10 ml anoxic PBS (pH 7.4) and did not use a PMA treatment. The small intestine contents for these mice were pelleted as described above.

## DNA isolation and 16S rRNA gene sequencing

We used the PowerSoil DNA isolation kit (Mo Bio Laboratories, Carlsbad, CA, USA) to extract DNA from these jejunal pellets frozen in 2 ml tubes containing 0.1 mm glass beads (Mo Bio Laboratories, Carlsbad, CA, USA). We eluted the DNA on the spin filter using 50 µl Solution C6 and stored the DNA at −20°C. We conducted blank extractions in parallel. We processed mouse fecal pellets in a similar manner.

We quantified DNA samples and blank extractions using the Quant-iT PicoGreen dsDNA Assay Kit. For each sample, we performed two 50 µl PCRs to amplify the V4 region of the 16S rRNA gene using primers 515F (f.c. 100 nM), Golay barcoded 806R (f.c. 100 nM) (*Caporaso et al., 2012*), 5 Prime Mix (Quanta Biosciences, CA, USA) or Classic++ Taq DNA Polymerase Master Mix (TONBO biosciences, CA, USA), and 25 ng of DNA. PCR conditions were 94°C for 3 min, 30 cycles of 94°C for 45 s, 50°C for 1 min, and 72°C for 1.5 min, followed by a final extension at 72°C for 10 min. Reactions were held at 4°C and stored at −20°C.

We combined the two 50 µl PCRs and purified DNA using Mag-Bind E-Z Pure (OMEGA Bio-tek, GA, USA) following the manufacturer's instructions and eluting with 35 µl TE. We measured DNA concentrations using PicoGreen. We pooled 100 ng of amplicon DNA from each sample together and sequenced the pool using the Illumina MiSeq 2 × 250 bp platform at the Cornell Biotechnology Resource Center Genomics Facility.

## 16S rRNA gene amplicon analysis

We processed, filtered, and analyzed the 16S rRNA gene amplicon data from all studies using QIIME 1.9.0 (*Caporaso et al., 2010*). Paired-end reads were joined using join_paired_ends.py running the fastq-join method and requiring at least 200 bp of sequence overlap. Joined reads were demultiplexed using split_libraries_fastq.py requiring a Phred quality cutoff of 25 to remove ambiguous barcodes and low quality reads. Reads were clustered into operational taxonomic units (OTUs) using open-reference OTU picking at 97% sequence identity to the Greengenes database version 13.8 (*DeSantis et al., 2006*). We focused our analyses on the two most abundant lactobacilli OTUs: OTU 692154 identified as *L. reuteri* and OTU 592160 as *L. johnsonii* as denoted by the Greengenes assignment. We confirmed these assignments by sequencing the full 16S rRNA gene of lactobacilli isolates (see below).

Except where noted, for all subsequent analyses, we rarified data to 40,000 sequences per sample. We calculated beta-diversity using the weighted UniFrac metric implemented in QIIME 1.9.0. We performed adonis (PERMANOVA) with 10,000 iterations and beta-diversity plots with the ordplot function using a t-distribution in the phyloseq package (*McMurdie and Holmes, 2013*). We

identified OTUs differentiating samples by first filtering OTU tables to only include those OTUs present in at least 25% of samples and with at least one sample having at least 100 counts of that OTU. To the filtered OTU tables, we applied a Kruskal-Wallis test with an FDR cutoff of 10% using the group_significance.py script in QIIME. We created heatmaps of OTUs passing with FDR < 0.1 using the make_otu_heatmap.py script in QIIME. To detect *L. reuteri* in the fecal pellets of F2 mice from Jackson Laboratories, samples with at least 10,000 sequences were used (sequencing depth was lower for the fecal pellets), and data were not rarefied so to maximize detection of *L. reuteri*.

## qPCR analysis of lactobacilli copy number altered by PMA treatment

We determined the copy numbers of the lactobacilli 16S rRNA gene and total Eubacterial 16S rRNA gene in the PMA and non-PMA treated jejunal aliquots by quantitative real-time PCR (qPCR) using the LightCycler 480 platform and the SYBR Green I Master kit (Roche Diagnostics Corporation, Indianapolis, IN, USA). We utilized the lactobacilli and Eubacterial primers described by *Oh et al., 2012*. PMA treatment reduces the total amount of DNA extracted by removing DNA from any dead cells. Thus, using the same mass of DNA for the PMA and non-PMA aliquots would result in quantifying copy numbers relative to the total amount of DNA assayed, similar to the relative abundances determined from the 16S rRNA gene sequencing. Therefore, we fixed the amount of DNA used for all non-PMA samples to 10 ng. Thus, 10 µl qPCRs consisted of 10 ng of DNA for the non-PMA aliquots and equal volume for the PMA aliquot, each qPCR primer at 500 nM, and 5 µl of SYBR Green I Master mix. Cycling conditions were 5 min at 95°C followed by 45 cycles consisting of 10 s at 95°C, 20 s at 56°C for the Eubacterial primers and 61°C for the lactobacilli primers, and 30 s at 72°C after which fluorescence from SYBR Green was read. Melting curve analysis was used to determine whether each qPCR reaction generated a unique product. Cycle threshold ($C_t$) values were calculated using the absolute quantification/2$^{nd}$ derivative max function available on the LightCycler 480 software. All reactions were run in triplicate, and the mean $C_t$ values were used in subsequent calculations.

To determine if the *Lactobacillus* population decreased due to the 18:2 gavage, we calculated the difference in lactobacilli copy number between the PMA (live-only cells) and non-PMA (total cells) aliquots relative to that for Eubacteria. That is,

$$\frac{2^{\Delta Ct\ Lacto(PMA-non.PMA)}}{2^{\Delta Ct\ Eubac(PMA-non.PMA)}}$$

If the *Lactobacillus* population is not affected by the 18:2 gavage, no difference should be observed between the saline and 18:2 gavage samples. Significance values between gavage groups were calculated using two-sample, two-tailed t-tests. Moreover, this ratio is expected to be close to one if lactobacilli were not specifically killed by the 18:2 gavage.

## Gavage ratio calculations

For each cage, we split the mice according to which gavage they received (18:2 or saline) and we took the mean of the rarefied sequence counts for OTU 692154 (*L. reuteri*). Then we calculated the $log_{10}$ of the ratio of the 18:2 mean rarefied sequence counts to the mean saline relative abundance sequence counts:

$$log_{10}\frac{mean\ 18:2\ counts\ per\ cage\ for\ OTU}{mean\ saline\ counts\ per\ cage\ for\ OTU}$$

## *L. reuteri* and *L. johnsonii* isolation from small intestine contents

We streaked the glycerol stocks of mouse ileum contents onto MRS agar plates. One or two colony morphologies were present on nearly all plates: lowly abundant bright cream, round colonies present on most plates, and abundant flatter, dull white colonies present on all plates. We determined the species identity of these colony morphologies by full length 16S rRNA gene sequencing using primers 27F (f.c. 1 nM) and 1391R (f.c. 1 nM) (*Turner et al., 1999*), 10 µl of Classic++ Hot Start Taq DNA Polymerase Master Mix (Tonbo Biosciences, CA, USA), and a small amount of a single bacterial colony in a 25 µl reaction. PCR conditions were 94°C for 3 min, 38 cycles of 94°C for 45 s, 50°C for 1 min, and 72°C for 1.5 min, followed by a final extension at 72°C for 10 min. We purified PCRs using Zymo DNA Clean and Concentrator−5 (Zymo Research, CA, USA) and submitted samples to Cornell University Institute of Biotechnology Sanger sequencing facility. Returned sequences were

assembled using Sequencher version 5.4.6 (DNA sequence analysis software, Gene Codes Corporation, Ann Arbor, MI, USA, http://www.genecodes.com) and aligned against National Center for Biotechnology Institute's nr database.

## Data deposition

The lactobacilli raw sequencing reads and the assembled genomes for strains LR0 and LJ0 are available under BioProject accession PRJNA376205 at National Center for Biotechnology Institute. The RAST genome annotations for these genomes are available in *Supplementary file 5* and *6*. The 16S rRNA gene amplicon data are available under the study accession PRJEB19690 at European Nucleotide Archive. Code to generate figures, mutational analysis pipelines, and relevant raw data are available at https://github.com/sdirienzi/Lactobacillus_soybeanoil (*Di Rienzi, 2017*; copy archived at https://github.com/elifesciences-publications/Lactobacillus_soybeanoil).

# Acknowledgements

We thank members of the Ley lab, as well as Jiyao Zhang, Donghao Wang, Andrew Clark, the staff of the Cornell Animal Facility, Jennifer Mosher, Sylvie Allen, Romano Miojevic, and Laura Ortiz-Velez for their assistance, helpful discussions, and insight.

# Additional information

### Competing interests

Ruth E Ley: Guest Reviewing Editor for eLife microbiome special issue. The other authors declare that no competing interests exist.

### Funding

| Funder | Grant reference number | Author |
| --- | --- | --- |
| NIH Office of the Director | DP2OD007444 | Ruth Emily Ley |
| Life Sciences Research Foundation | Eli & Edythe Broad Fellow | Sara C Di Rienzi |
| Max-Planck-Gesellschaft | Open-access funding | Ruth Emily Ley |

The funders had no role in study design, data collection and interpretation, or the decision to submit the work for publication.

### Author contributions

Sara C Di Rienzi, Conceptualization, Data curation, Formal analysis, Funding acquisition, Investigation, Methodology, Writing—original draft, Project administration, Writing—review and editing; Juliet Jacobson, Elizabeth A Kennedy, Mary E Bell, Investigation; Qiaojuan Shi, Investigation, Project administration; Jillian L Waters, Peter Lawrence, J Thomas Brenna, Investigation, Methodology; Robert A Britton, Resources, Investigation, Methodology, Writing—review and editing; Jens Walter, Resources, Methodology, Writing—review and editing; Ruth E Ley, Conceptualization, Resources, Supervision, Funding acquisition, Investigation, Writing—original draft, Project administration, Writing—review and editing

### Author ORCIDs

Sara C Di Rienzi (iD) http://orcid.org/0000-0002-6188-663X
Jens Walter (iD) http://orcid.org/0000-0003-1754-172X
Ruth E Ley (iD) http://orcid.org/0000-0002-9087-1672

### Ethics

Animal experimentation: All animal experimental procedures were reviewed and approved by the Institutional Animal Care and Usage Committee of Cornell University protocol 2010-0065.

**Decision letter and Author response**
Decision letter https://doi.org/10.7554/eLife.32581.029
Author response https://doi.org/10.7554/eLife.32581.030

## Additional files

**Supplementary files**

• Supplementary file 1. *L. reuteri* strains isolated from various hosts. Host, strain name, country of origin, clade, and site of isolation on the human body (if applicable) are given.
DOI: https://doi.org/10.7554/eLife.32581.021

• Supplementary file 2. Lactobacilli in vitro population sequencing. Tab 'SequencingCoverage' gives information on the sequencing run, barcode, number of sequences obtained, and estimated genomic coverage. Tab 'AncestorGenomes' gives information on the assembled LJ0 and LR0 genomes.
DOI: https://doi.org/10.7554/eLife.32581.022

• Supplementary file 3. Filtered and unfiltered mutations in the *L. reuteri in vitro* population and HF diet isolate sequencing data. Tab 'Key' describes the information in the subsequent tabs. Tab 'All' shows all variants passing filtering by GATK. Tab 'Filtered' shows variants filtered to exclude alleles in genes annotated with 'mobile element protein', 'transposase', 'phage', or 'RNA', alleles at less than 10% frequency, and alleles at frequency greater than 0.5 in the aligned ancestor reads against the reference genome. Tab 'Handchecked' shows variants passing previous filtering and confirmed manually in IGV. For *L. reuteri* two additional tabs are included: 'LRHF only' shows variants only found in the *L. reuteri* isolate from a mouse on the HF SBO diet. 'Populations only' tab shows variants only found in the in vitro evolution assay. Genomic details are taken from the RAST annotation of the ancestor genome. Other columns are taken from the GATK vcf file. For allele variants falling in intergenic regions, the surrounding genes are listed in HitGene and HitChrom, HitStrand, HitStart, HitEnd, HitDNA, and HitProtein are listed as 'NA'.
DOI: https://doi.org/10.7554/eLife.32581.023

• Supplementary file 4. Filtered and unfiltered mutations in the *L. johnsonii in vitro* population sequencing data. Tabs and details are the same as for *Supplementary file 3*.
DOI: https://doi.org/10.7554/eLife.32581.024

• Supplementary file 5. RAST annotation for the assembled LR0 genome.
DOI: https://doi.org/10.7554/eLife.32581.025

• Supplementary file 6. RAST annotation for the assembled LJ0 genome.
DOI: https://doi.org/10.7554/eLife.32581.026

• Transparent reporting form
DOI: https://doi.org/10.7554/eLife.32581.027

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
