## [Decision Letter]

Thank you for sending your article entitled "Resilience of small intestinal lactobacilli to the toxicity of soybean oil fatty acids" for peer review at *eLife*. Your article has been favorably evaluated by Wendy Garrett (Senior Editor) and three peer reviewers.

Summary:

Di Rienzi et al. propose that the types of lipids consumed on a Western diet have led to a decline in the prevalence of *Lactobacillus*. First, they perform in vitro experiments on cultured *Lactobacillus*, confirming that linoleic acid (LA) inhibits the in vitro growth of *L. reuteri*. Comparison of strains from multiple host animals reveals variation in sensitivity. Passaging of sensitive strains led to the evolution of resistance and genome sequencing revealed potential candidate genes responsible. Finally, they perform a dietary supplementation experiment in mice fed a low-fat and high-fat diet which did not result in a marked impact of LA on gut bacterial abundance. The authors use this data to propose that something about the gut environment protects sensitive strains from toxic dietary components.

Impression:

Di Rienzi et al., present an interesting set of experiments testing the hypothesis that commensal microbes have developed resistance to certain bacteriocidal components of the Western Diet. This hypothesis is highly relevant to the study of microbiome shifts in response to Western Dietary changes and can be applied to many aspects of the diet beyond fatty acids. In this regard, this study sets up a nice in vitro model for systematically testing these concepts. Furthermore, this study asks an interesting and important question: Do dietary components deleteriously affect the microbiome?

Essential revisions:

More information is needed about the animals used, currently described as a combination of 48 mice purchased from Jackson Labs, 24 mice purchased from Taconic, and 26 F2 generation mice bred in-house. How were facility-specific differences in baseline composition controlled for? Were there any attempts at normalization through spreading bedding and fecal material across cages or other methodologies?

Analysis of the starting microbiome from each animal should be included to allow within-subject comparisons. The mouse's starting level of *Lactobacillus* may be responsible for the numbers post-gavage.

A description of the mutations found in the sequenced isolates (LR2-1 and LJ4107) is missing. Population-level mutations are described, but it would be good to know how many mutations were present in the isolates and if they are similar to those found in the populations, since the resistance profiles of these isolates was specifically measured. It could be important to know whether resistance can arise through single mutations/pathways, or require multiple mutations/pathways, as this may help predict the likelihood of resistance occurring in vivo.

This study states and presumes that 28 μg per ml of linoleic acid is representative of in vivo gut concentrations. However, neither of the cited references in the first paragraph of the subsection “*L. reuteri* strains show variable resistance to 18:2 in vitro” is a quantification of in vivo lumenal mammalian linoleic acid concentrations – both are culture-based studies showing that *Lactobacilli* are inhibited by similar concentrations of fatty acids used by the authors. Without confidence that this in vivo estimate is accurate, one cannot rule out the alternative that the lack of in vivo inhibition observed was due to in vivo concentrations not reaching MIC due to some combination of the speeds in in vivo of lipid breakdown speed, absorption and metabolism by other microbes. For this reason, the authors must do one or more of the following: (i) better explain and support the basis for the estimate they provide; (ii) provide data on which this estimate is based; or (iii) provide one or more references in which luminal linoleic acid is quantified in humans or mice.

The impact of this paper would be substantially improved by either identifying the mechanism of in vitro resistance or even better, by determining at least one mechanism through which bacteria might be protected in vivo. The latter is likely beyond the scope of the 2-month resubmission period, but the former seems feasible. Options might include knocking out a few putative resistance genes (for example, FabT) in the parent strain, including complementation controls.

---

## [Author Response]

Essential revisions:More information is needed about the animals used, currently described as a combination of 48 mice purchased from Jackson Labs, 24 mice purchased from Taconic, and 26 F2 generation mice bred in-house. How were facility-specific differences in baseline composition controlled for? Were there any attempts at normalization through spreading bedding and fecal material across cages or other methodologies?

The different sets animals were maintained at different times. Our goal was to replicate our findings regarding *Lactobacilli* with a completely different set of mice with a different base microbiota to ensure our findings were not specific to a given facility, but were general to host-associated *Lactobacilli*. Therefore, we wanted to maximize the differences between mouse sets. To clarify this point, we have added the following to the Materials and methods section:

“The three different sets of mice were maintained at distinct time periods with the goal of ensuring our findings were not specific to a given base-microbiota.”

Analysis of the starting microbiome from each animal should be included to allow within-subject comparisons. The mouse's starting level of Lactobacillus may be responsible for the numbers post-gavage.

Unfortunately, this is not possible. Our study focuses on the jejunum, which can only be sampled destructively. To this end, we included in our study a saline gavage as a control. The validity of this control as representing the starting microbiome is shown in the observation that the 18:2 gavage and saline gavage microbiomes are indistinguishable if the total cell populations (live and dead cells) are considered. Mice within the same cage have more similar microbiomes than mice in other cages on the same diet. The saline gavage was applied to half of the mice in each cage, and we included in our statistical analyses cage as a component; thus, we were comparing the saline and 18:2 gavaged mice within the same cage.

A description of the mutations found in the sequenced isolates (LR2-1 and LJ4107) is missing. Population-level mutations are described, but it would be good to know how many mutations were present in the isolates and if they are similar to those found in the populations, since the resistance profiles of these isolates was specifically measured. It could be important to know whether resistance can arise through single mutations/pathways, or require multiple mutations/pathways, as this may help predict the likelihood of resistance occurring in vivo.

Thank you for pointing out this omission. We have added this information to the main text. In short, both LR2-1 and LJ4107 have all of the high frequency mutations present in their source population as well as one additional mutation, which are present at frequencies 45% and 39% in their respective source populations.

“The isolate LR2-1 contained both of the mutations present at high frequencies in the total LR2 population as well as an additional mutation in a hypothetical protein, which was present in the LR2 population at 45% (Supplementary file 3). Similarly, LJ41072 had all of the high frequency mutations present in its source population (LJ4) and one additional mutation in LafA, which was mutated in 39% of the LJ4 population (Supplementary file 4).”

This study states and presumes that 28 μg per ml of linoleic acid is representative of in vivo gut concentrations. However, neither of the cited references in the first paragraph of the subsection “L. reuteri strains show variable resistance to 18:2 in vitro” is a quantification of in vivo lumenal mammalian linoleic acid concentrations – both are culture-based studies showing that Lactobacilli are inhibited by similar concentrations of fatty acids used by the authors. Without confidence that this in vivo estimate is accurate, one cannot rule out the alternative that the lack of in vivo inhibition observed was due to in vivo concentrations not reaching MIC due to some combination of the speeds in in vivo of lipid breakdown speed, absorption and metabolism by other microbes. For this reason, the authors must do one or more of the following: (i) better explain and support the basis for the estimate they provide; (ii) provide data on which this estimate is based; or (iii) provide one or more references in which luminal linoleic acid is quantified in humans or mice.

To estimate the concentration of linoleic acid present in the small intestine, we took into account how much food our mice consumed and the linoleic acid composition of the diet. This calculation has been added to the manuscript:

“The mice in this study consumed on average 2.7 grams of mouse food per day. […] For the 7% by weight SBO diet (16% by calorie LF diet), 11 to 28 μg/ml 18:2 will pass in a transit time.”

The impact of this paper would be substantially improved by either identifying the mechanism of in vitro resistance or even better, by determining at least one mechanism through which bacteria might be protected in vivo. The latter is likely beyond the scope of the 2-month resubmission period, but the former seems feasible. Options might include knocking out a few putative resistance genes (for example, FabT) in the parent strain, including complementation controls.

With the help of Robert Britton, now added to the paper as a co-author, we have generated two of the FabT mutations and the mutation in the hydrolase gene observed in our *L. reuteri* populations in the fatty acid sensitive strain PTA 6475. This strain is amenable to recombineering. Both of the FabT mutations improved the fatty acid resistance of the strain, but the hydrolase mutation did not. These data are presented in the last two paragraphs of the subsection “Evolved 18:2 resistance is associated with mutations in lipid-related, acid stress, and cell membrane/wall genes”, Figure 3—figure supplement 2B, a new Materials and methods subsection “Generation of *L. reuteri* mutants”, and Figure 3—source data 1. The other genes mutated in the *L. reuteri* populations are not present in PTA 6475.